# Gamification as a health education strategy of adolescents at school: Protocol for a systematic review and meta-analysis

**Thais Teixeira dos Santos**[1,2©], **Manuel Pardo Ríos**[3], **Gidyenne Christine Bandeira Silva de Medeiros**[1,2,4], **Ádala Nayana de Sousa Mata**[2,5], **Danyllo do Nascimento Silva Junior**[1,2], **Daniel Martínez Guillen**[3], **Grasiela Piuvezam**[1,2,6©]*

**1** Post-Graduation Program in Public Health, Federal University of Rio Grande do Norte, Natal, Brazil, **2** Sistematic Review and Meta-analisys Laboratory (Lab-Sys/CNPq), Federal University of Rio Grande do Norte (UFRN), Natal, Brazil, **3** Faculty of Nursing, Catholic University of Murcia, Murcia, Spain, **4** Department of Nutrition, Federal University of Rio Grande do Norte, Natal, Rio Grande do Norte, Brazil, **5** Multicampi School of Medical Sciences of Rio Grande do Norte (EMCM), Federal University of Rio Grande do Norte, Caicó, Brazil, **6** Department of Public Health, Federal University of Rio Grande do Norte (UFRN-Brazil), Natal, Brazil

© These authors contributed equally to this work.
* gpiuvezam@yahoo.com.br

## Abstract

The objective of the study is to present a systematic review and meta-analysis protocol for evaluating the effects of health education gamification-based interventions, on health parameters (food consumption, sleep quality and physical activity) of adolescent students. This protocol was registered on the International Prospective Register of Systematic Reviews (PROSPERO) database (CRD42022373833). The search will be performed in the following databases: MEDLINE, Embase, Scopus, ERIC, ScienceDirect, Web of Science, Cochrane, LILACS, APA, and ADOLEC. Intervention studies (clinical trials - randomized or non-randomized) and quasi-experimental studies will be included. The risk of bias will be assessed using the Risk of Bias in Non-randomized Studies of Interventions tool for randomized controlled trials, non- randomized controlled trials and quasi-experimental trials. Two independent researchers will conduct all assessments, and any disagreements will be consulted with a third reviewer. Data analysis and synthesis will be analyzed using RevMan 5.4.1 software. We will conduct the study in accordance with the Preferred Reporting Items for Systematic Reviews and Meta-Analyses Protocols (PRISMA-P) guideline. The review will summarize the current evidence on gamification health education changes in parameters related to the health of adolescents. Gamification has been used to verify the increase in adherence to healthy habits or the development of health-related skills, but there are still few results for the adolescent population. We expect that the systematic review could indicate strategies with gamification interventions and also determine how these strategies can improve health parameters of adolescent students, and they will be available as a reference for these interventions.

**Data Availability Statement:** Deidentified research data will be made publicly available when the study is completed and published.

**Funding:** This study was financed in part by the Coordenação de Aperfeiçoamento de Pessoal de Nível Superior-Brasil (CAPES) (finance code: 001), and in part by the Ignacio H Larramendi Research Grant granted by the Mapfre Foundation in 2022 to UCAM-Catholic University of Murcia for the 'PECES' research project to MPR and his research team (finance code: N/A).

**Competing interests:** The authors have declared that no competing interests exist.

# Introduction

Adolescence is defined by the World Health Organization [1] as the period between 10 and 19 years of age. This phase is characterized by the gradual transition from childhood to adulthood with the appearance of signs of puberty, which comprise important physiological, psychological, and social changes [2].

Due to all of these changes, there is an increasing interest for in-depth studies on of this phase, especially with regard to health aspects. Also, this population has increased in recent years, and it is already well known that the health-related habits during the period of adolescence will have a lasting impact on the adults they will become, and the children of the next generation [3].

Children and adolescents may have difficulties in adapting to new, healthier lifestyles, and therefore, efforts need to be made to apply new approaches aimed at the successful adherence to these new habits [4]. In this perspective, in many intervention strategies implemented in studies aimed at promoting health among adolescents, techniques have been adopted based on immersive technologies, such as gamification, by bringing this population closer to technology and electronic devices [5, 6].

According to Deterding et al. [7], 'gamification' or 'game-based learning' corresponds to the "use of game design elements to improve academic performance", such as learning attitudes, behaviors and results. Furthermore, some authors [8–11] clearly define gamification as the use of game elements in non-gaming contexts. The evaluation of gamified resources is one of the great debates of today. It is not yet clear how to measure these interventions and whether there is also an improvement in learning outcomes, despite there being an increase in student motivation. Future research should clarify mechanisms underlying gamified educational interventions and explore theories that could explain the effects of these interventions on learning outcomes, using well-defined control groups, in a longitudinal manner [12].

The use of gamification in health education promotes dialogue between the trained team and intervention participants, with active engagement and the promotion of participant reflection in a relaxing learning environment [13]. Furthermore, the use of active games by children and adolescents can help to promote healthy habits [14], such as improving diet, eating habits and body composition [15], improving the sleep quality and symptoms of sleep interruption due to the COVID-19 pandemic [16], for example, and in the practice of physical activity [17].

It is important to highlight that positive results have been obtained with patients diagnosed with illnesses such as diabetes [13], when seeking to promote playful education [18], and general change of behavior for healthier life habits in children [6].

Thus, the objective of the study is to present a protocol that will assist in conducting a systematic review and meta-analysis to evaluate the effects of health education gamification-based interventions on health parameters (food consumption, sleep quality and physical activity) of adolescent students.

# Methods and analysis

## Study registration and reporting

This protocol was registered on the International Prospective Register of Systematic Reviews (PROSPERO) database on November 29th, 2022 (CRD42022373833). The protocol is based on the Preferred Reporting Items for Systematic Reviews and Meta-Analyses Protocols (PRISMA-P) guidelines [19] (S1 Checklist). The final report will be developed following PRISMA [20] and the Cochrane Handbook for Systematic Reviews of Interventions [21], and any

changes to the protocol will be described in the Methods section. The planned start and end dates registered are January 10th, 2024, and September 10[th], 2024, respectively.

The review questions guiding this proposal are: what are the main intervention strategies with gamification techniques for the health education of adolescent students? What are the effects of using gamification/game-based interventions on sleep quality parameters of adolescent students? What are the effects of using gamification/game-based interventions on the food consumption parameters of adolescent students? What are the effects of using gamification/game-based interventions on the physical activity parameters of adolescent students?

## Eligibility criteria

**Types of study.** Intervention studies (clinical trials - randomized or non-randomized) and quasi-experimental studies that describe interventions using gamification as a health education strategy with adolescent students, will be included. The latter must necessarily have a control group.

**Participants.** Only studies whose population is adolescents will be included. For this investigation, the definition of "adolescents" by the World Health Organization [22] will be considered, which refers to a person aged between 10 and 19 years old.

**Intervention.** Studies in which interventions with gamification for health education were carried out will be included.

**Compares.** We will consider studies with a control group that compare the intervention with the no gamification intervention group.

**Outcome measures.** The main expected outcome is changes in parameters related to the health of adolescents, such as: food consumption parameters (eating frequency; eating habits; biological, psychological, and psychological and sociocultural factors related to food), sleep quality (sleep habits, sleep disorders), and physical activity (aerobic, anaerobic and sports modalities; health benefits) that support or explain the effect of interventions based on the use of gamification in the health education of adolescents. The studies to be included must present at least one of the expected outcomes.

**Exclusion criteria.** We will exclude studies in which interventions were exclusively applied to certain diseases/pathologies and when the population includes young people (from 10 to 24 years old) or children without an analysis of the subgroup of adolescents. Studies without a control group or intervention, or when the undergoing interventions do not use gamification will also be excluded. Studies that evaluate only nutrients and not the food will not be included. Adolescents will not be excluded due to the level of education (high school or college or technical school), as long as they meet the other eligibility criteria.

## Search strategies

The reviews will be divided into three thematic areas: food consumption, sleep quality, and physical activity. For each thematic area, reviewers should follow the following steps: apply the inclusion and exclusion criteria when searching the databases by reading the titles and abstracts; apply the eligibility criteria after reading the articles selected in the initial stage in full; assess the methodological quality and the risk of bias of the articles included in the previous step and perform a quantitative synthesis of data from selected articles (meta-analysis).

**Electronic search.** The search will be performed in the following databases: MEDLINE/ PubMed, Embase, Scopus, ERIC - Education Resources Information Center, ScienceDirect, Web of Science, LILACS–Latin American and Caribbean Health Sciences Literature, American Psychological Association PsycINFO, and ADOLEC.

A free combination of Medical Subject Heading (MeSH) terms and Keywords will be performed. The search string will be defined considering the following items: participants (adolescents; students), intervention (gamification; school; health education), outcomes (diet, food and nutrition; dairy products; sleep quality; dyssomnias; sleep disorders, intrinsic; sleep hygiene; exercise; sedentary behavior), and study design (clinical trial, intervention, observational).

The search terms for composing the strings will be combined with specific filters in each database (S1 File). There will be no restrictions on publication date or language in the searches.

**Additional search.** To ensure the scope of this research, we will complement the searches with a manual search in the reference lists of the retrieved studies or relevant reviews that are related to the topic.

## Screening procedure

Two systematic reviews will be conducted: one addressing the main gamification strategies used in health education interventions, and another addressing the main effects of these interventions on the dimensions of food consumption, sleep, and physical activity of adolescents.

For all identified studies, at least two authors will independently select and review titles and abstracts using the Rayyan® application for systematic reviews [23]. Articles that meet the inclusion criteria will be ordered for a full review. Any disagreement will be resolved by discussion with a third reviewer. A manual search will be performed if any relevant studies are found using the defined search strategies. All researchers will then review the full text of all eligible studies. The information on the phases of the selection process will be described through a Preferred Reporting Items for Systematic Reviews and Meta-Analyses Protocols diagram (PRISMA Flow diagram) [20].

## Data extraction

Two reviewers will extract the following information from the relevant studies selected: publication identity, participant characteristics, control group, intervention characteristics,

**Table 1. Data extraction.**

| DATA TO BE EXTRACTED | ITEM |
| --- | --- |
| Publication ID | Title, first author, publication year, country, study name, population. |
| Study design | Randomized Controlled Trials. Non-Randomized Controlled Trials. Quasi-experimental studies. |
| Participants' characteristics | Sex. Age. Sample size. |
| Control group | No intervention. Other NOT Game-based Intervention |
| Intervention characteristics | Duration of intervention. Follow-up period. Intervention description. Game-based learning approach. Social factors associated. |
| Outcomes measurements | Food consumption. Sleep quality. Physical activity. |
| Analysis methods | Statistical methods used. Quantitative synthesis. |

outcome measures, and analysis methods. Some examples of data extraction are available in Table 1. Any disagreement will be resolved through a discussion and review of the article, and a third researcher will be consulted.

## Data analysis

**Risk of bias and quality assessment.**   Two independent researchers will carry out the evaluation, and in case of doubts or discrepancies, a third researcher will be consulted. The methodological quality of randomized clinical trials will be assessed using Version 2 of the Cochrane Risk of Bias (ROB 2) tool for randomized clinical trials [24].

For non-randomized, before-after controlled studies, the risk of bias will be assessed using the Risk of Bias in Non-randomized Studies of Interventions (ROBINS-I) tool, which was developed to assess the risk of bias in the results of non-randomized studies and quasi-experimental studies with a control group that compare the effects of two or more interventions on the health of the study population, by classifying the risk of bias as low, high, or unclear [25].

The overall strength of evidence for each outcome will be analyzed using the Grading of Recommendations Assessment, Development and Evaluation (GRADE) tool [26]. To calculate the inter-rater reliability, we will use the kappa coefficient.

**Statistical analysis.**   A narrative approach will be used to summarize the effectiveness of the interventions regarding food consumption, sleep quality, and physical activity. The use of gamification as a health education strategy in intervention studies will be analyzed separately. If the studies are methodologically homogeneous, a meta-analysis will be conducted.

Review Manager (RevMan V5.3.3) software will be used for the data analysis. Heterogeneity between assay results will be evaluated by performing a standard chi-square test with a significance level of 0.05. To assess heterogeneity, the $I^2$ statistic will be calculated, corresponding to a quantitative measure of inconsistency between studies. A value of 0% indicates that no heterogeneity was observed, while $I^2$ values of 50% indicate a moderate level, and 75% or higher indicate a substantial level of heterogeneity.

If possible, funnel plots will be used to assess the presence of potential reporting biases. A linear regression will be performed to evaluate the asymmetry of the funnel plot. If the studies are heterogeneous, a narrative summary will be carried out.

**Missing data.**   In the case of interesting data missing or being unclear, the research team will try to contact the corresponding author by email, phone or correspondence. If this communication is unsuccessful, the data from the analysis will be excluded. This will be covered in the Discussion section, to describe the possible impact of missing data.

**Subgroup analysis.**   If sufficient data are available, the following subgroup analyses will be performed: specific details of the interventions (e.g. methodological strategy, components, and duration), and research setting (family participation, socioeconomic conditions).

## Patient and public involvement

Patients and/or the public were not involved in the design, performance, reporting, or dissemination plans of this research. Patient consent for publication is not applicable.

## Ethics and dissemination

Ethical approval and human consent are not required because this is a protocol for a systematic review, and only secondary data will be used. Findings will be published in a peer-reviewed journal and presented at conferences. In case of any changes in this protocol, the protocol will be updated in the International Prospective Register of Systematic Reviews (PROSPERO) website, and the modifications will be explained in the final report of this review.

## Discussion

The purpose of this investigation is to carry out systematic reviews and meta-analyses to evaluate the main effects of interventions with gamification in the dimensions of food consumption, sleep quality, and physical activity of adolescent students.

The gaming market is increasing around the world, with valuations of US$175.8 billion in the year 2021, and the expectation that it will exceed US$200 billion by 2023 [27]. Brazil is the largest market in Latin America [28] for many products, with videogames one of them. With the expansion of this industry, the applications of games will not only be for entertainment purposes, but also gamification of education.

### Implications

Gamification can be applied in several areas of health and education dimensions. In the educational context, gamification is used to offer students the possibility of learning autonomously [29].

The use of gamification can also make long-term activities more pleasurable, with positive cognitive, social, and emotional results [30]. In the teaching-learning process, the use of gamification has increased as a strategy to encourage student motivation and involvement in this process, and to influence the social aspect [31]. However, it is important to highlight that to achieve the full potential of the application of gamification, it is important to deeply understand its mechanisms and the psychological effects of its use [32].

Still in this context, Lee and Hammer [33] describe the importance of the dimension of challenging oneself when using gamification techniques in learning, as this makes students develop new skills. In addition, gamification brings out emotions in the players, such as pride, joy, optimism, curiosity, and even frustration. Finally, the authors make it clear that there is also a stimulus towards the social component, as the student's results and achievements are made known to other people, which could further encourage students to improve.

It is the application of different game mechanisms that guarantees the development of cognitive skills, such as achieving greater concentration, problem solving capacity, and a sense of orientation towards an end goal [34], producing brain responses to extrinsic stimuli received [35]. Thus, the use of game-based learning techniques can be an attractive alternative for promoting health education in adolescents.

During an initial analysis of the literature, a diversity was found in the application of gamification depending on the object of study. Therefore, it is possible to find research involving the application of gamification in health promotion and disease prevention [36, 37] in teaching health professionals [38, 39], in teaching during the COVID-19 pandemic [40], and in school learning [41].

Some reviews document the positive effect of interventions using gamification to increase the knowledge of children and adolescents regarding habits related to healthier eating and nutrition [14, 42, 43]. In addition to these, it is also possible to find systematic reviews that attest to the positive effects of gamification in encouraging greater adherence to physical activity in isolation [44], and even in association with healthy eating habits [45].

Another study that evaluated the use of gamification in applications related to health habits reported that 97% of the evaluated tools used such techniques, regarding the practice of physical activity and weight loss, down to 11% when the focus was on sleep [46]. Regarding this last variable, an alternative for promoting sleep hygiene is the use of serious game techniques [47], which seek effects on emotional, psychological and social well-being [16].

In addition, it can be stated that the proposal to use active games strategically follows the logic that it is about the union between fun and activities aimed at health, through prevention,

promotion, and improvement of its parameters [48], with increased energy expenditure and reduced sedentary behavior [6].

## Limitations

We may find some limitations with this review. These may be: the lack of systematic reviews that evaluate the dimensions of food consumption, sleep quality, and physical activity of the adolescent population, associating them with a gamification intervention; the difficulty in finding studies that evaluate food consumption and not the nutritional composition of the diet of adolescents; the studies on health education of adolescents may involve children [49] and/or young adults [50], moving away from our proposal to focus on adolescence; the variety of study types included can also generate heterogeneity between studies, which could compromise the quantitative summary of the results (meta-analysis).

Despite these limitations, the elucidation of the main gamification techniques used in the health education of adolescents, and the main findings related to the effects of these applications in the promotion of the health of this population, is expected from the development of the systemic review from this protocol article. As an example, the following can be cited: a greater adherence to healthy eating habits, and the practice of physical activity, improvement in sleep quality, and subsequent reduction in sleep disorders and sedentary behavior.

## Supporting information

**S1 Checklist. PRISMA-P (Preferred Reporting Items for Systematic review and Meta-Analysis Protocols) 2015 checklist: Recommended items to address in a systematic review protocol\*.**
(DOC)

**S1 File. Draft of search strategy.**
(DOC)

## Author Contributions

**Conceptualization:** Thais Teixeira dos Santos, Gidyenne Christine Bandeira Silva de Medeiros, Ádala Nayana de Sousa Mata, Grasiela Piuvezam.

**Data curation:** Thais Teixeira dos Santos, Manuel Pardo Ríos, Gidyenne Christine Bandeira Silva de Medeiros, Ádala Nayana de Sousa Mata, Danyllo do Nascimento Silva Junior, Daniel Martínez Guillen, Grasiela Piuvezam.

**Formal analysis:** Thais Teixeira dos Santos, Manuel Pardo Ríos, Gidyenne Christine Bandeira Silva de Medeiros, Ádala Nayana de Sousa Mata, Danyllo do Nascimento Silva Junior, Daniel Martínez Guillen, Grasiela Piuvezam.

**Investigation:** Thais Teixeira dos Santos, Danyllo do Nascimento Silva Junior.

**Methodology:** Thais Teixeira dos Santos, Manuel Pardo Ríos, Gidyenne Christine Bandeira Silva de Medeiros, Ádala Nayana de Sousa Mata, Danyllo do Nascimento Silva Junior, Daniel Martínez Guillen, Grasiela Piuvezam.

**Project administration:** Thais Teixeira dos Santos, Gidyenne Christine Bandeira Silva de Medeiros, Grasiela Piuvezam.

**Supervision:** Thais Teixeira dos Santos, Gidyenne Christine Bandeira Silva de Medeiros, Ádala Nayana de Sousa Mata, Grasiela Piuvezam.

**Writing – original draft:** Thais Teixeira dos Santos.

**Writing – review & editing:** Thais Teixeira dos Santos, Gidyenne Christine Bandeira Silva de Medeiros, Ádala Nayana de Sousa Mata, Grasiela Piuvezam.

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
