## [Decision Letter · Decision Letter 0]

2 May 2023

PONE-D-23-04396Gamification as school adolescents’ health education strategy: protocol for a systematic review and meta-analysisPLOS ONE

Dear Dr. Piuvezam,

Thank you for submitting your manuscript to PLOS ONE. After careful consideration, we feel that it has merit but does not fully meet PLOS ONE’s publication criteria as it currently stands. Therefore, we invite you to submit a revised version of the manuscript that addresses the points raised during the review process.

The reviewers have provided feedback on the manuscript for further improvement. To enhance the quality of the manuscript, attention should be given to the following four key points in any subsequent revision:

1. Language and Abbreviations: The authors should improve the English writing and correct any errors that may be present. Additionally, abbreviations must be defined upon first appearance in the text. Non-standard abbreviations should be avoided unless they appear at least three times in the text, and their use should be kept to a minimum.

2. Introduction: The introduction should be improved by justifying how gamification can be associated with the parameters that will be measured.

3. Methodology: The methodology should be reviewed to ensure that the risk of bias in the studies included is measured appropriately. The exclusion criteria should be clarified, and the research questions proposed should be clearly stated. The information should also be presented in a logical sequence.

4. References: The references should be reviewed to ensure that they correspond to what is intended to be cited.

In addition, authors should adhere to the following formatting guidelines:

1. PLOS One guidelines for Study Protocols (https://journals.plos.org/plosone/s/submission-guidelines#loc-study-protocols), including the use of the PRISMA-P checklist for systematic review protocols (http://www.prisma-statement.org/Extensions/Protocols).

2. Although optional, authors are encouraged to register with OSF and provide the registration number in the Materials and Methods section.

3. The PLOS One Study Protocol Article Template and an OSF discipline or study-specific template should be used as a guide.

4. The discussion should include any issues involved in performing the study that are not covered in other sections, such as limitations of the study design, dissemination plans, and how amendments to the study will be dealt with.

We look forward to receiving your revised manuscript.

Kind regards,

Delfina Fernandes Hlashwayo, M.Sc.

Academic Editor

PLOS ONE

2. Please note that we require the PRISMA-P checklist as supplementary file to be provided with protocols for systematic reviews. Therefore, please remove the standard PRISMA checklist from your submission and provide a completed PRISMA-P checklist as supplementary file instead. For further information please refer to: https://www.prisma-statement.org/Extensions/Protocols. Thank you for your attention. We look forward to hearing from you.

3. We note that your article has been submitted as a "Registered Report Protocol" article type. Please note that your submission is not suitable for the article type "Registered Report Protocol". The "Registered Report Protocol" article type is only suitable for proposals of studies that have not yet started, and which will undergo editorial assessment per the Registered Reports framework. When resubmitting your manuscript, we ask that you update your article type to "Study Protocol" in the online submission form. Please note that some fields in the submission form, particularly in the "Additional Information" field, will have been reset with this change, so please go through your submission in full to ensure that all information is accurate and complete when resubmitting your manuscript.

“This study was partially financed by the Coordenação de Aperfeiçoamento de Pessoal de Nível Superior - Brasil (CAPES) - Finance Code 001”

Reviewers' comments:

Reviewer's Responses to Questions

**Comments to the Author**

1. Does the manuscript provide a valid rationale for the proposed study, with clearly identified and justified research questions?

Reviewer #1: No

Reviewer #2: Yes

2. Is the protocol technically sound and planned in a manner that will lead to a meaningful outcome and allow testing the stated hypotheses?

Reviewer #1: Yes

Reviewer #2: Partly

3. Is the methodology feasible and described in sufficient detail to allow the work to be replicable?

Reviewer #1: Yes

Reviewer #2: Yes

4. Have the authors described where all data underlying the findings will be made available when the study is complete?

Reviewer #1: Yes

Reviewer #2: Yes

5. Is the manuscript presented in an intelligible fashion and written in standard English?

Reviewer #1: Yes

Reviewer #2: No

6. Review Comments to the Author

You may also provide optional suggestions and comments to authors that they might find helpful in planning their study.

Reviewer #1: Dear Authors,

Thank you for writing this protocol.

Please find below my comments

1. Your title indicates that you want to do a quantitative systematic review but in your abstract, you have mentioned the inclusion of both quantitative and qualitative study. If all you are including in your study are quantitative studies, then your title is fine. But if you are including both quantitative and qualitative studies than your title needs to reflect that it is a mixed methods systematic review protocol.

2. What is RX, please say what that means in the abstract.

3. You have mentioned the use of Risk of Bias in Non-randomised studies for RCTs, please check that and correct.

Introduction

4. From your introduction, could you please provide more evidence/ examples on how gamification is associated with the three main health parameters you are considering. (Food consumption, sleep quality and physical activity).

5. Could you also state in your introduction, that this review hasn’t been done. This implies you checking different databases and ascertaining that the systematic review does not exist.

Methods 2.1

6. Rather than have a protocol that encompasses two systematic reviews, why don’t you do two separate systematic reviews. 1. A systematic review on the content of gamification and 2.) the systematic review, which you are currently proposing. Otherwise please adjust the topic of your systematic review to reflect what you want to achieve.

7. Please check the spelling of ‘quase’-experimental. What is RQ2, RQ3 and RQ4. They have not been stated anywhere else in the document, hence, it is better that you state them out clearly rather than use RQ1, RQ2 and RQ3.

Methods 2.2

8. Line 142 – You may consider using the PICO format (Participants, Intervention, Comparator and Outcome)

9. Please check and correct Subgroup ‘analyzes’

Methods 2.3

10. I am not sure why you have thematic areas here since you have already identified them as your outcomes previously.

Discussion

11. I would rather that some of the evidence that you have given in the discussion be presented more concisely in the introduction. As it is a protocol which is to describe what you want to do; it is not necessary to have a discussion part yet.

Reviewer #2: This paper provides a protocol for a systematic review and meta-analysis that intends to evaluate the effects of gamified interventions in health education on health risk factors (food consumption, sleep quality and physical activity) in adolescents.

The protocol is registered with PROSPERO and was developed according to the PRISMA statement guidelines. Four research questions are specified and these will be addressed by two systematic reviews and a meta-analysis. As part of the discussion and justification for conducting the study, some indication is provided of the scope of papers available to complete the review and meta-analysis, but not the results of such. The paper may be useful to researchers because it provides a detailed example of how to conduct a review and meta-analysis.

Recommended courses of action

The author appears to be writing in English as a second language as the choice of words and phrases in not as expected in common use. For example, use of the word ‘public’ as in Lines 92,101,142, is unusual and may be better understood if replaced by a word such as ‘demographic’ or ‘age-group’. Also, the word ‘parameters’ in Line 120, may be better replaced by ‘behaviours’. As a general comment the English grammar in parts of the paper would benefit from an editorial review. Eg Line 158, It will be excluded studies in which:

Reference is made in Lines 129-133 to the research questions RQ1 to RQ4 but these are not directly defined. The research questions are specified in Lines182-189 but without the shortened name, and it would improve clarity if the abbreviations were used in these lines along with the definition eg Line 183 ……..RQ1 What are the main intervention strategies with gamification…..?

Line 201: Why are dairy foods singled out for inclusion in the MeSH and Keywords when other food groups important for health are not included eg fruit and vegetables.

Lines 161- 163, “Young people (from 10 to 24 years old) will be excluded ………….no longer fit as adolescents (over 19 years old);” Would these studies be excluded if data available separately for the age groups of interest?

Is the reference provided in Lines 506-508, the correct one for citation 25 on Line 261?

Line 275 Please state what criteria would make it possible to use funnel plots to assess reporting bias?

7. PLOS authors have the option to publish the peer review history of their article (what does this mean?). If published, this will include your full peer review and any attached files.

Reviewer #1: No

Reviewer #2: No

---

## [Author Response · Author response to Decision Letter 0]

29 May 2023

To Editor and reviwers 

Plos One

Subject: Submission of the revised Study Protocol article entitled “Gamification as a health education strategy of adolescents at school: protocol for a systematic review and meta-analysis”

I confirm that we have made the adjustments indicated by Plos One academic editor and reviewer(s) in the PLOS ONE Decision: Revision required [PONE-D-23-04396] - [EMID:9192f11600e35ee3] email. The following are responses to the review requests:

Dear Dr. Piuvezam,

Thank you for submitting your manuscript to PLOS ONE. After careful consideration, we feel that it has merit but does not fully meet PLOS ONE’s publication criteria as it currently stands. Therefore, we invite you to submit a revised version of the manuscript that addresses the points raised during the review process.

The reviewers have provided feedback on the manuscript for further improvement. To enhance the quality of the manuscript, attention should be given to the following four key points in any subsequent revision:

• Language and Abbreviations: The authors should improve the English writing and correct any errors that may be present. Additionally, abbreviations must be defined upon first appearance in the text. Non-standard abbreviations should be avoided unless they appear at least three times in the text, and their use should be kept to a minimum.

• We improved the English writing and corrected any errors that were present. Abbreviations were kept to a minimum.

• Introduction: The introduction should be improved by justifying how gamification can be associated with the parameters that will be measured.

• The introduction was improved as requested.

3. Methodology: The methodology should be reviewed to ensure that the risk of bias in the studies included is measured appropriately. The exclusion criteria should be clarified, and the research questions proposed should be clearly stated. The information should also be presented in a logical sequence.

• The methodology was reviewed to ensure that the risk of bias in the studies included is measured appropriately, as well we clarified the exclusion criteria and the research questions proposed.

4. References: The references should be reviewed to ensure that they correspond to what is intended to be cited.

• We also reviewed the references to ensure that they correspond to what is intended to be cited.

• Our manuscript meets PLOS ONE's style requirements, and the PLOS ONE style templates were used;

2. Please note that we require the PRISMA-P checklist as supplementary file to be provided with protocols for systematic reviews. Therefore, please remove the standard PRISMA checklist from your submission and provide a completed PRISMA-P checklist as supplementary file instead. For further information please refer to: https://www.prisma-statement.org/Extensions/Protocols. Thank you for your attention. We look forward to hearing from you.

• We provided the PRISMA-P checklist as supplementary file.

3. We note that your article has been submitted as a "Registered Report Protocol" article type. Please note that your submission is not suitable for the article type "Registered Report Protocol". The "Registered Report Protocol" article type is only suitable for proposals of studies that have not yet started, and which will undergo editorial assessment per the Registered Reports framework. When resubmitting your manuscript, we ask that you update your article type to "Study Protocol" in the online submission form. Please note that some fields in the submission form, particularly in the "Additional Information" field, will have been reset with this change, so please go through your submission in full to ensure that all information is accurate and complete when resubmitting your manuscript.

• When resubmitting our manuscript, we updated our article type to "Study Protocol" in the online submission form.

“This study was partially financed by the Coordenação de Aperfeiçoamento de Pessoal de Nível Superior - Brasil (CAPES) - Finance Code 001”

• We updated our Funding statement to: This study was financed in part by the Coordenação de Aperfeiçoamento de Pessoal de Nível Superior - Brasil (CAPES) - Finance Code 001; and in part by the Ignacio H. Larramendi Research Grant granted by the Mapfre Foundation in 2022 to UCAM - Catholic University of Murcia for the "PECES" research project to Dr. Manuel Pardo Ríos and his research team. The funders had no role in study design, data collection and analysis, decision to publish, or preparation of the manuscript;

• We updated our Data Availability statement to: All relevant data are within the paper and its Supporting Information files. No datasets were generated or analyzed during the current study;

Reviewers' comments:

Reviewer's Responses to Questions

Comments to the Author

1. Does the manuscript provide a valid rationale for the proposed study, with clearly identified and justified research questions?

Reviewer #1: No

Reviewer #2: Yes

Kind regards,

Grasiela Piuvezam

Lab-SYS (Systematic Review and Meta-Analysis Laboratory-CNPq)

Department of Public Health

Department of Postgraduate in Public Health, 

Federal University of Rio Grande do Norte (UFRN)

Campus Lagoa Nova, CEP 59078-970, PO Box 1524

Natal/RN, Brazil

---

## [Editor Report · Decision Letter 1]

22 Jun 2023

PONE-D-23-04396R1Gamification as a health education strategy of adolescents at school: protocol for a systematic review and meta-analysisPLOS ONE

Dear Dr. Piuvezam,

Thank you for submitting your manuscript to PLOS ONE. After careful consideration, we feel that it has merit but does not fully meet PLOS ONE’s publication criteria as it currently stands. Therefore, we invite you to submit a revised version of the manuscript that addresses the points raised during the review process.

We would like to express our appreciation for your efforts in addressing the editor's questions and meeting the journal's requirements in the recent revision of your manuscript.

While reviewing the revised manuscript, we noticed that the "Response to Reviewers" section does not include the specific comments and concerns raised by the two reviewers, as well as how those comments were addressed. In order to provide a comprehensive evaluation and reach a final decision, it is crucial for us to have access to this information. Therefore, we kindly request that you revise the "Response to Reviewers" section to include the comments from both reviewers and clearly indicate how each comment was addressed in the manuscript. 

Including the reviewers' comments and the corresponding responses will help us make an informed decision and provide valuable feedback to the reviewers. Additionally, it will facilitate a transparent and thorough evaluation of the manuscript.

We look forward to receiving your revised manuscript.

Kind regards,

Delfina Fernandes Hlashwayo, M.Sc.

Academic Editor

PLOS ONE
---

## [Author Response · Author response to Decision Letter 1]

3 Jul 2023

July 3rd, 2023

Rebuttal Letter for PLOS ONE

Dear Oriel Jerome Delas Alas Vida, 

 Thank you for considering our manuscript for publication at PLOS ONE. We are certain of the credibility and commitment that PLOS ONE has to quality in science, and we hope that the modifications made in our manuscript fully meet the publication criteria. 

 We have sure that the suggestions add quality and clarity to our paper. We list at the end of this letter the answers for each one of them.

 Once again, thank you for considering our manuscript and we are at your disposal for any further information. 

Sincerely,

The authors. 

Authors’ responses to the Editor (Email June 30th, 2023)

1. Please review your reference list to ensure that it is complete and correct. If you have cited papers that have been retracted, please include the rationale for doing so in the manuscript text or remove these references and replace them with relevant current references. Any changes to the reference list should be mentioned in the rebuttal letter that accompanies your revised manuscript. If you need to cite a retracted article, indicate the article’s retracted status in the References list and also include a citation and full reference for the retraction notice.

We have reviewed the reference list and confirmed that we do not cite any retracted articles. The adjustments made to the reference list are highlighted and referred to the required formatting. All cited references refer to corresponding citations throughout the manuscript text and they are complete and correct.

Authors’ responses to the Editor (Email May 2nd, 2023)

2. Language and Abbreviations: The authors should improve the English writing and correct any errors that may be present. Additionally, abbreviations must be defined upon first appearance in the text. Non-standard abbreviations should be avoided unless they appear at least three times in the text, and their use should be kept to a minimum.

We improved the English writing and corrected any errors that were present. We add the certificate of professional translation during the submission of the revised article. Abbreviations were kept to a minimum. 

3. Introduction: The introduction should be improved by justifying how gamification can be associated with the parameters that will be measured.

The introduction was improved as requested. 

4. Methodology: The methodology should be reviewed to ensure that the risk of bias in the studies included is measured appropriately. The exclusion criteria should be clarified, and the research questions proposed should be clearly stated. The information should also be presented in a logical sequence.

The methodology was reviewed to ensure that the risk of bias in the studies included is measured appropriately, as well we clarified the exclusion criteria and the research questions proposed.

5. References: The references should be reviewed to ensure that they correspond to what is intended to be cited.

We also reviewed the references to ensure that they correspond to what is intended to be cited. 

Authors’ responses to Journal requirements (Email May 2nd, 2023)

6. Please ensure that your manuscript meets PLOS ONE's style requirements, including those for file naming. The PLOS ONE style templates can be found at https://journals.plos.org/plosone/s/file?id=wjVg/PLOSOne_formatting_sample_main_body.pdf and https://journals.plos.org/plosone/s/file?id=ba62/PLOSOne_formatting_sample_title_authors_affiliations.pdf

Our manuscript meets PLOS ONE's style requirements, and the PLOS ONE style templates were used.

7. Please note that we require the PRISMA-P checklist as supplementary file to be provided with protocols for systematic reviews. Therefore, please remove the standard PRISMA checklist from your submission and provide a completed PRISMA-P checklist as supplementary file instead. For further information please refer to: https://www.prisma-statement.org/Extensions/Protocols. Thank you for your attention. We look forward to hearing from you.

We provided the PRISMA-P checklist as supplementary file.

8. We note that your article has been submitted as a "Registered Report Protocol" article type. Please note that your submission is not suitable for the article type "Registered Report Protocol". The "Registered Report Protocol" article type is only suitable for proposals of studies that have not yet started, and which will undergo editorial assessment per the Registered Reports framework. When resubmitting your manuscript, we ask that you update your article type to "Study Protocol" in the online submission form. Please note that some fields in the submission form, particularly in the "Additional Information" field, will have been reset with this change, so please go through your submission in full to ensure that all information is accurate and complete when resubmitting your manuscript.

When resubmitting our manuscript, we updated our article type to "Study Protocol" in the online submission form.

9. Thank you for stating the following financial disclosure:

“This study was partially financed by the Coordenação de Aperfeiçoamento de Pessoal de Nível Superior - Brasil (CAPES) - Finance Code 001”

We updated our Funding statement to: This study was financed in part by the Coordenação de Aperfeiçoamento de Pessoal de Nível Superior - Brasil (CAPES) - Finance Code 001; and in part by the Ignacio H. Larramendi Research Grant granted by the Mapfre Foundation in 2022 to UCAM - Catholic University of Murcia for the "PECES" research project to Dr. Manuel Pardo Ríos and his research team. The funders had no role in study design, data collection and analysis, decision to publish, or preparation of the manuscript.

10. We note that you have stated that you will provide repository information for your data at acceptance. Should your manuscript be accepted for publication, we will hold it until you provide the relevant accession numbers or DOIs necessary to access your data. If you wish to make changes to your Data Availability statement, please describe these changes in your cover letter and we will update your Data Availability statement to reflect the information you provide.

We updated our Data Availability statement to: All relevant data are within the paper and its Supporting Information files. No datasets were generated or analyzed during the current study.

Authors’ Responses to Reviewers' comments (Email May 2nd, 2023)

11. Does the manuscript provide a valid rationale for the proposed study, with clearly identified and justified research questions?

Reviewer #1: No

Reviewer #2: Yes

Considering the importance of making clear the justification for our study, as well as the research questions, we decided to add this information more clearly in the introduction of our manuscript. In addition, after the suggestions received, we made changes to highlight the innovative character of our study for an existing valid academic problem, as well as the contribution of the construction of this study to science. We hope we were able to clarify this information after reviewing the manuscript.

---

## [Decision Letter · Decision Letter 2]

31 Jul 2023

PONE-D-23-04396R2Gamification as a health education strategy of adolescents at school: protocol for a systematic review and meta-analysisPLOS ONE

Dear Dr. Piuvezam,

Thank you for submitting your manuscript to PLOS ONE. After careful consideration, we feel that it has merit but does not fully meet PLOS ONE’s publication criteria as it currently stands. Therefore, we invite you to submit a revised version of the manuscript that addresses the points raised during the review process. Please submit your revised manuscript by Sep 14 2023 11:59PM. If you will need more time than this to complete your revisions, please reply to this message or contact the journal office at plosone@plos.org. Please include the following items when submitting your revised manuscript:A rebuttal letter that responds to each point raised by the academic editor and reviewer(s). You should upload this letter as a separate file labeled 'Response to Reviewers'.A marked-up copy of your manuscript that highlights changes made to the original version. You should upload this as a separate file labeled 'Revised Manuscript with Track Changes'.An unmarked version of your revised paper without tracked changes. You should upload this as a separate file labeled 'Manuscript'.If applicable, we recommend that you deposit your laboratory protocols in protocols.io to enhance the reproducibility of your results. Protocols.io assigns your protocol its own identifier (DOI) so that it can be cited independently in the future. For instructions see: https://journals.plos.org/plosone/s/submission-guidelines#loc-laboratory-protocols. Additionally, PLOS ONE offers an option for publishing peer-reviewed Lab Protocol articles, which describe protocols hosted on protocols.io. Read more information on sharing protocols at https://plos.org/protocols?utm_medium=editorial-email&utm_source=authorletters&utm_campaign=protocols.

We look forward to receiving your revised manuscript.

Kind regards,

Delfina Fernandes Hlashwayo, M.Sc.

Academic Editor

PLOS ONE

Journal Requirements:

**Additional Editor Comments:**

The reviewers have provided feedback on the manuscript to enhance its quality. In addition to the reviewer’s comments, the authors are advised to consider the following four key points for any subsequent revisions:

To enhance the ease of reference, we kindly request that you present the PRISMA-P checklist using page numbers instead of line numbers.While the strategy for Embase is present in the supplementary file, we recommend adding the search strategies for the following databases as well: MEDLINE, Scopus, ERIC, ScienceDirect, Web of Science, Cochrane, LILACS, APA, and ADOLEC.The addition of a translation certificate as a supplementary file is not required. We kindly ask that you exclude it from the submission. Similarly, the funding disclosure should not be included as a supplementary file.

Reviewers' comments:

Reviewer's Responses to Questions

**Comments to the Author**

1. Does the manuscript provide a valid rationale for the proposed study, with clearly identified and justified research questions?

Reviewer #3: Yes

Reviewer #4: Partly

2. Is the protocol technically sound and planned in a manner that will lead to a meaningful outcome and allow testing the stated hypotheses?

Reviewer #3: Yes

Reviewer #4: Partly

3. Is the methodology feasible and described in sufficient detail to allow the work to be replicable?

Reviewer #3: Yes

Reviewer #4: No

4. Have the authors described where all data underlying the findings will be made available when the study is complete?

Reviewer #3: Yes

Reviewer #4: Yes

5. Is the manuscript presented in an intelligible fashion and written in standard English?

Reviewer #3: Yes

Reviewer #4: No

6. Review Comments to the Author

You may also provide optional suggestions and comments to authors that they might find helpful in planning their study.

Reviewer #3: The authors carefully responded to all requests from reviewers.

In general, the study's proposal is innovative and will bring contributions to teaching and learning, as well as future research and university extension actions for adolescents.

The introduction is well delimited and now well justified in relation to the purpose of the protocol. The authors propose a search in several databases, which may help to overcome a study limitation of finding research to compose the study, as long as the theme is relatively new and directed only to adolescents.

Some notes:

In eligibility criteria: It must be clear in the protocol whether students up to high school or college or technical students will be included (In some countries there are technical education courses). As a result of in the first semesters of college or technical courses most are composed by teenagers.

The suggestion is that the approach be taken regardless of the level of schooling in progress, as data to be extracted and analyzed.

Reviewer #4: It is interesting work but not entirely clear to the reader. The abstract was well written but I could not understand it fully despite reading the main manuscript 2-3 times.

7. PLOS authors have the option to publish the peer review history of their article (what does this mean?). If published, this will include your full peer review and any attached files.

Reviewer #3: **Yes: **Flaviana Vely Mendonça Vieira

Reviewer #4: No

---

## [Author Response · Author response to Decision Letter 2]

13 Sep 2023

September 14th, 2023

Rebuttal Letter for PLOS ONE [PONE-D-23-04396R2] -[EMID:12c7612585d2c131]

Dear M.Sc. Delfina Fernandes Hlashwayo, 

 Thank you for considering our manuscript for publication at PLOS ONE. We are certain of the credibility and commitment that PLOS ONE has to quality in science, and we hope that the modifications made in our manuscript fully meet the publication criteria. 

 We have read the reviewers' and editor’s comments carefully, answering all questions in this letter and made the requested changes in the revised manuscript. We are sure that the suggestions add quality and clarity to our paper. We hope the revised manuscript is now suitable for publication at PLOS ONE.

 Once again, thank you for considering our manuscript and we are at your disposal for any further information. 

Best regards,

The authors. 

Authors’ responses to the Editor

1. To enhance the ease of reference, we kindly request that you present the PRISMA-P checklist using page numbers instead of line numbers.

We appreciate your availability to revise our manuscript. We attended to your request and update PRISMA-P using page numbers.

2. While the strategy for Embase is present in the supplementary file, we recommend adding the search strategies for the following databases as well: MEDLINE, Scopus, ERIC, ScienceDirect, Web of Science, Cochrane, LILACS, APA, and ADOLEC.

We thank you for your considerations. PRISMA-P Statement recommends, we have to “specify all databases, registers, websites, organisations, reference lists, and other sources searched or consulted to identify studies. Specify the date when each source was last searched or consulted”.

We confirm that we will follow the Preferred Reporting Items for Systematic Reviews and Meta-Analyses Protocols (PRISMA-P) guidelines in developing our systematic review, but we have chosen to make it available in this protocol article, as a supplementary file, the precise and full search strategies for only one database (EMBASE) including any filters and limits used, as a way of guarding against a possible misuse of our search strategies while we are developing the systematic review.

3. The addition of a translation certificate as a supplementary file is not required. We kindly ask that you exclude it from the submission. Similarly, the funding disclosure should not be included as a supplementary file.

We thank you for this recommendation. We will exclude both translation certificate and funding disclosure from the submission.

Authors’ Responses to Reviewers' comments:

1. Does the manuscript provide a valid rationale for the proposed study, with clearly identified and justified research questions?

Reviewer #3: Yes

Reviewer #4: Partly

We appreciate your availability to revise our manuscript and all your comments. Considering the innovative nature of using gamification as a health education strategy for schoolchildren, we guided our research questions according to the dimensions to be explored in the construction of the review: food consumption, sleep and physical activity. The research questions, in addition to being in the registered PRISMA-P protocol, are fully described in the manuscript methodology, more specifically on page 5, lines 94-11, and they have been constructed in accordance with the justification presented in the Introduction section.

2. Is the protocol technically sound and planned in a manner that will lead to a meaningful outcome and allow testing the stated hypotheses?

Reviewer #3: Yes

Reviewer #4: Partly

We thank you for your revision. As described in the manuscript, the protocol is based on the Preferred Reporting Items for Systematic Reviews and Meta-Analyses Protocols (PRISMA-P) guidelines, which can be found as supplementary material S1. We will follow all the necessary steps to develop the systematic review according to the described protocol, which was methodologically developed to allow us to test the hypotheses and answer the questions proposed in our study.

We hope that the results presented in the systematic review that will be developed based on this protocol can guide the creation of new health education tools for school adolescents using immersive technologies.

3. Is the methodology feasible and described in sufficient detail to allow the work to be replicable?

Reviewer #3: Yes

Reviewer #4: No

We appreciate your revision. We confirm that the protocol is based on the Preferred Reporting Items for Systematic Reviews and Meta-Analyses Protocols (PRISMA-P) guidelines and the final report will be developed following PRISMA and the Cochrane Handbook for Systematic Reviews of Interventions. 

Therefore, we confirm that this protocol has enough details for another researcher to reproduce the review and analyses. Furthermore, the protocol contains a description of all eligibility criteria, the selection steps that will be followed, the data that will be extracted and the possible analyzes to be carried out. Therefore, it is a protocol that can be reproduced.

4. Have the authors described where all data underlying the findings will be made available when the study is complete?

Reviewer #3: Yes

Reviewer #4: Yes

We thank you for your revision. All information related to the construction of the article to be published was made available in the files added in the submission.

5. Is the manuscript presented in an intelligible fashion and written in standard English?

Reviewer #3: Yes

Reviewer #4: No

Thank you for your comments. We reviewed the translation of our article with a professional translator.

6. Review Comments to the Author

You may also provide optional suggestions and comments to authors that they might find helpful in planning their study.

Reviewer #3: The authors carefully responded to all requests from reviewers.

In general, the study's proposal is innovative and will bring contributions to teaching and learning, as well as future research and university extension actions for adolescents.

The introduction is well delimited and now well justified in relation to the purpose of the protocol. The authors propose a search in several databases, which may help to overcome a study limitation of finding research to compose the study, as long as the theme is relatively new and directed only to adolescents.

Some notes:

In eligibility criteria: It must be clear in the protocol whether students up to high school or college or technical students will be included (In some countries there are technical education courses). As a result of in the first semesters of college or technical courses most are composed by teenagers.

The suggestion is that the approach be taken regardless of the level of schooling in progress, as data to be extracted and analyzed.

Reviewer #4: It is interesting work but not entirely clear to the reader. The abstract was well written but I could not understand it fully despite reading the main manuscript 2-3 times.

We thank you for all the comments and evaluations presented. We always do our best to respond to suggestions and increase the quality of our work. 

The amount of database is extensive so that our search is more comprehensive, and we have a better chance of finding articles that can answer our research questions.

We thank you for the suggestion regarding the study locations of the adolescents included in the studies and made an adjustment to the eligibility criteria (highlighted on page 6 of the manuscript)

To make our introduction more understandable, we made some adjustments.

7. PLOS authors have the option to publish the peer review history of their article (what does this mean?). If published, this will include your full peer review and any attached files.

Do you want your identity to be public for this peer review? For information about this choice, including consent withdrawal, please see our Privacy Policy.

Reviewer #3: Yes: Flaviana Vely Mendonça Vieira

Reviewer #4: No

We are grateful for all the contributions presented by both reviewers to improve the construction of our article.

---

## [Decision Letter · Decision Letter 3]

31 Oct 2023

PONE-D-23-04396R3Gamification as a health education strategy of adolescents at school: protocol for a systematic review and meta-analysis

PLOS ONE

Dear Dr. Piuvezam,

Thank you for submitting your manuscript to PLOS ONE. After careful consideration, we feel that it has merit but does not fully meet PLOS ONE’s publication criteria as it currently stands. Therefore, we invite you to submit a revised version of the manuscript that addresses the points raised during the review process.

To enhance the overall quality and accuracy of your manuscript, please take into consideration the following key points:

Ensure to check for any spelling, grammar, or phrasing errors.When referring to the level of education (high school, college, or technical school), please avoid using the term "place of study."Review the planned start and end dates, as the protocol has not yet been published. The registered start and end dates are January 10th, 2023, and September 8th, 2023, which have passed.Kindly revise the text in the methodology section where it is written in the past tense (line 129).Please ensure the correct naming of Supplementary Files.In line 201, please provide the complete name of the program. Additionally, in line 202, consider using the correct symbol for the chi-square test or write it out in full.Please avoid defining the included population as "population used" (line 287).Please submit your revised manuscript by Dec 15 2023 11:59PM. If you will need more time than this to complete your revisions, please reply to this message or contact the journal office at plosone@plos.org. Please include the following items when submitting your revised manuscript:A rebuttal letter that responds to each point raised by the academic editor and reviewer(s). You should upload this letter as a separate file labeled 'Response to Reviewers'.A marked-up copy of your manuscript that highlights changes made to the original version. You should upload this as a separate file labeled 'Revised Manuscript with Track Changes'.An unmarked version of your revised paper without tracked changes. You should upload this as a separate file labeled 'Manuscript'.If applicable, we recommend that you deposit your laboratory protocols in protocols.io to enhance the reproducibility of your results. Protocols.io assigns your protocol its own identifier (DOI) so that it can be cited independently in the future. For instructions see: https://journals.plos.org/plosone/s/submission-guidelines#loc-laboratory-protocols. Additionally, PLOS ONE offers an option for publishing peer-reviewed Lab Protocol articles, which describe protocols hosted on protocols.io. Read more information on sharing protocols at https://plos.org/protocols?utm_medium=editorial-email&utm_source=authorletters&utm_campaign=protocols.

We look forward to receiving your revised manuscript.

Kind regards,

Delfina Fernandes Hlashwayo, Ph.D.

Academic Editor

PLOS ONE

Journal Requirements:

Reviewers' comments:

Reviewer's Responses to Questions

**Comments to the Author**

1. Does the manuscript provide a valid rationale for the proposed study, with clearly identified and justified research questions?

Reviewer #3: Yes

2. Is the protocol technically sound and planned in a manner that will lead to a meaningful outcome and allow testing the stated hypotheses?

Reviewer #3: Yes

3. Is the methodology feasible and described in sufficient detail to allow the work to be replicable?

Reviewer #3: Yes

4. Have the authors described where all data underlying the findings will be made available when the study is complete?

Reviewer #3: Yes

5. Is the manuscript presented in an intelligible fashion and written in standard English?

Reviewer #3: Yes

6. Review Comments to the Author

You may also provide optional suggestions and comments to authors that they might find helpful in planning their study.

Reviewer #3: I appreciate the opportunity to evaluate this study, which makes an important contribution to the area.

The authors made the indicated adjustments.

7. PLOS authors have the option to publish the peer review history of their article (what does this mean?). If published, this will include your full peer review and any attached files.

Reviewer #3: No

---

## [Author Response · Author response to Decision Letter 3]

6 Nov 2023

November 7th, 2023

Rebuttal Letter for PLOS ONE [PONE-D-23-04396R3] 

Dear M.Sc. Delfina Fernandes Hlashwayo, 

 Thank you for considering our manuscript for publication at PLOS ONE. We are certain of the credibility and commitment that PLOS ONE has to quality in science, and we hope that the modifications made in our manuscript fully meet the publication criteria. 

 We have read the reviewer's and editor’s comments carefully, answering all questions in this letter and made the requested changes in the revised manuscript. We are sure that the suggestions add quality and clarity to our paper. We hope the revised manuscript is now suitable for publication at PLOS ONE.

 Once again, thank you for considering our manuscript and we are at your disposal for any further information. 

Best regards,

The authors. 

Authors’ responses to the Editor

1. Ensure to check for any spelling, grammar, or phrasing errors.

We appreciate your availability to revise our manuscript. We attended to your request, and we revised all the text.

2. When referring to the level of education (high school, college, or technical school), please avoid using the term "place of study."

We thank you for your considerations and we have made the changes.

3. Review the planned start and end dates, as the protocol has not yet been published. The registered start and end dates are January 10th, 2023, and September 8th, 2023, which have passed.

We thank you for this recommendation. We reviewed the planned start and end dates in the protocol.

4. Kindly revise the text in the methodology section where it is written in the past tense (line 129).

We appreciate your considerations and we have reviewed the text.

5. Please ensure the correct naming of Supplementary Files.

We thank you for this recommendation. We reviewed the names of Supplementary Files.

6. In line 201, please provide the complete name of the program. Additionally, in line 202, consider using the correct symbol for the chi-square test or write it out in full.

We appreciate your considerations. We provided the information required.

7. Please avoid defining the included population as "population used" (line 287).

We are grateful for your recommendations. All suggestions were very important to improve the quality of our study.

Authors’ Responses to Reviewers' comments:

1. Does the manuscript provide a valid rationale for the proposed study, with clearly identified and justified research questions? The research question outlined is expected to address a valid academic problem or topic and contribute to the base of knowledge in the field.

Reviewer #3: Yes

We thank you for all the comments and evaluations presented. We always do our best to respond to suggestions and increase the quality of our work. 

2. Is the protocol technically sound and planned in a manner that will lead to a meaningful outcome and allow testing the stated hypotheses? The manuscript should describe the methods in sufficient detail to prevent undisclosed flexibility in the experimental procedure or analysis pipeline, including sufficient outcome-neutral conditions (e.g. necessary controls, absence of floor or ceiling effects) to test the proposed hypotheses and a statistical power analysis where applicable. As there may be aspects of the methodology and analysis which can only be refined once the work is undertaken, authors should outline potential assumptions and explicitly describe what aspects of the proposed analyses, if any, are exploratory.

Reviewer #3: Yes

We appreciate your revision.

3. Is the methodology feasible and described in sufficient detail to allow the work to be replicable? Descriptions of methods and materials in the protocol should be reported in sufficient detail for another researcher to reproduce all experiments and analyses. The protocol should describe the appropriate controls, sample size calculations, and replication needed to ensure that the data are robust and reproducible.

Reviewer #3: Yes

We appreciate your revision, and we confirm that this protocol can be reproduced.

4. Have the authors described where all data underlying the findings will be made available when the study is complete? The PLOS Data policy requires authors to make all data underlying the findings described in their manuscript fully available without restriction, with rare exception, at the time of publication. The data should be provided as part of the manuscript or its supporting information, or deposited to a public repository. For example, in addition to summary statistics, the data points behind means, medians and variance measures should be available. If there are restrictions on publicly sharing data—e.g. participant privacy or use of data from a third party—those must be specified.

Reviewer #3: Yes

We are grateful for your revision. All information related to the construction of the article to be published was made available in the files added in the submission.

5. Is the manuscript presented in an intelligible fashion and written in standard English? PLOS ONE does not copyedit accepted manuscripts, so the language in submitted articles must be clear, correct, and unambiguous. Any typographical or grammatical errors should be corrected at revision, so please note any specific errors here.

Reviewer #3: Yes

Thank you for your comments. We reviewed the translation of our article with a professional translator.

6. Review Comments to the Author. Please use the space provided to explain your answers to the questions above and, if applicable, provide comments about issues authors must address before this protocol can be accepted for publication. You may also include additional comments for the author, including concerns about research or publication ethics.

You may also provide optional suggestions and comments to authors that they might find helpful in planning their study. (Please upload your review as an attachment if it exceeds 20,000 characters)

Reviewer #3: I appreciate the opportunity to evaluate this study, which makes an important contribution to the area.

The authors made the indicated adjustments.

We appreciate your availability to revise our manuscript and all your comments. We hope that the results presented in the systematic review that will be developed based on this protocol can be an instrument on health education for school adolescents.

7. PLOS authors have the option to publish the peer review history of their article (what does this mean?). If published, this will include your full peer review and any attached files. Do you want your identity to be public for this peer review? For information about this choice, including consent withdrawal, please see our Privacy Policy.

Reviewer #3: No

We are grateful for all your contributions presented to improve the construction of our article.

---

## [Editor Report · Decision Letter 4]

13 Nov 2023

Gamification as a health education strategy of adolescents at school: protocol for a systematic review and meta-analysis

PONE-D-23-04396R4

Dear Dr. Piuvezam,

We’re pleased to inform you that your manuscript has been judged scientifically suitable for publication and will be formally accepted for publication once it meets all outstanding technical requirements.

Kind regards,

Delfina Fernandes Hlashwayo, Ph.D.

Academic Editor

PLOS ONE

Additional Editor Comments (optional):

Please ensure the thorough elimination of any grammatical errors throughout the manuscript

Line 63: Instead of "the authors" I suggest using "some authors" or another appropriate expression.

There is a repetition of the phrase "systematic review" in lines 81-82. Please adjusting to avoid redundancy.

Line 116: I suggest using "compares" in the plural for greater accuracy.

Line 130: Please review the wording in this line for clarity and precision.

The sentence in the first paragraph of the limitations section is excessively long. Please revise and split it for improved flow.
---

## [Editor Report · Acceptance letter]

22 Nov 2023

PONE-D-23-04396R4 

Gamification as a health education strategy of adolescents at school: protocol for a systematic review and meta-analysis 

Dear Dr. Piuvezam:

I'm pleased to inform you that your manuscript has been deemed suitable for publication in PLOS ONE. Congratulations! Your manuscript is now with our production department. 

Kind regards, 

on behalf of

Ms. Delfina Fernandes Hlashwayo 

Academic Editor

PLOS ONE